# Fabrication of Celecoxib PVP Microparticles Stabilized by Gelucire 48/16 via Electrospraying for Enhanced Anti-Inflammatory Action

**DOI:** 10.3390/ph16020258

**Published:** 2023-02-08

**Authors:** Samar Zuhair Alshawwa, Thanaa A. El-Masry, Engy Elekhnawy, Hadil Faris Alotaibi, Al-Sayed Sallam, Dalia H. Abdelkader

**Affiliations:** 1Department of Pharmaceutical Sciences, College of Pharmacy, Princess Nourah bint Abdulrahman University, P.O. Box 84428, Riyadh 11671, Saudi Arabia; 2Pharmacology and Toxicology Department, Faculty of Pharmacy, Tanta University, Tanta 31527, Egypt; 3Pharmaceutical Microbiology Department, Faculty of Pharmacy, Tanta University, Tanta 31527, Egypt; 4Al-Taqaddom Pharmaceutical Industries, Amman 11947, Jordan; 5Pharmaceutical Technology Department, Faculty of Pharmacy, Tanta University, Tanta 31527, Egypt

**Keywords:** electrospraying, enhanced oral bioavailability, micellar solubilization, carrageenan-induced inflammation, cyclooxygenase 2, gene expression, proinflammatory cytokine

## Abstract

Electrospraying (ES) technology is considered an efficient micro/nanoparticle fabrication technique with controlled dimensions and diverse morphology. Gelurice^®^ 48/16 (GLR) has been employed to stabilize the aqueous dispersion of Celecoxib (CXB) for enhancing its solubility and oral bioavailability. Our formula is composed of CXB loaded in polyvinylpyllodine (PVP) stabilized with GLR to formulate microparticles (MPs) (CXB-GLR-PVP MPs). CXB-GLR-PVP MPs display excellent in vitro properties regarding particle size (548 ± 10.23 nm), zeta potential (−20.21 ± 2.45 mV), and drug loading (DL, 1.98 ± 0.059 mg per 10 mg MPs). CXB-GLR-PVP MPs showed a significant (*p* < 0.05) higher % cumulative release after ten minutes (50.31 ± 4.36) compared to free CXB (10.63 ± 2.89). CXB exhibited good dispersibility, proved by X-ray diffractometry (XRD), adequate compatibility of all components, confirmed by Fourier-Transform Infrared Spectroscopy (FTIR), and spherical geometry as revealed in scanning electron microscopy (SEM). Concerning our anti-inflammatory study, there was a significant decrease in the scores of the inflammatory markers’ immunostaining in the CXB-GLR-PVP MPs treated group. Also, the amounts of the oxidative stress biomarkers, as well as mRNA expression of interleukins (IL-1β and IL-6), considerably declined (*p* < 0.05) in CXB-GLR-PVP MPs treated group alongside an enhancement in the histological features was revealed. CXB-GLR-PVP MPs is an up-and-coming delivery system that could be elucidated in future clinical investigations.

## 1. Introduction

Celecoxib (CXB) is a Class II candidate that belongs to nonsteroidal anti-inflammatory drugs. It has superior anti-inflammatory, antipyretic, and analgesic potential [1]. Its pharmacological properties are attributed to hampering the activity of the COX-2 enzyme and decreasing the synthesis of prostaglandin (PG) [2]. Although it has several magic pharmacological features mentioned above, it possesses several formulation obstacles hindering its optimal delivery after oral administration. Poor aqueous solubility of CXB leads to a low dissolution rate and hence, low oral bioavailability [3]. So, improving the aqueous solubility of CXB in an urgent formulation is a necessity enabling it to exert an immediate anti-inflammatory action. The process of inflammation is a protective response of the human cells toward various inducers, such as infections, tissue injury, and certain chemical compounds. During this process, the induction of some reactions occurs via certain mediators, which leads to the occurrence of inflammation [4]. These mediators, like interleukin-6 (IL-6), interleukin 1 beta (IL-1β), and tumor necrosis factor-alpha (TNF-α), start and provoke the inflammatory process [5]. The enzymes that are involved in the inflammatory reaction, like cyclooxygenase 2 (COX-2), have the ability to increase the release of prostaglandins (PGs) during the inflammatory process [6].

Diverse formulations like micro-/nano-emulsions, liposomes, mixed micelles, and solid dispersion (SD) systems were previously utilized to enhance the solubility of poorly soluble agents [7,8,9,10,11]. Recently, efficient size reduction techniques like the fabrication of nano/microparticles via electrospraying (ES) are also included [12]. ES is a potential technology for the creation of microparticles (MPs) ideal for various therapeutic applications because it is a facile, uncomplicated, and direct method. Furthermore, it is crucial for the biomedical industry because of its adaptability to encapsulate pharmaceuticals in MPs and manage the processing of biodegradable polymeric MPs after tight adjustment of process variables, including feed rate, operating voltage, the distance separating the needle tip and collecting drum, etc. [12].

In our study, we utilized Gelurice^®^ 48/16 (GLR) as a micellar solubilizer enhancing the aqueous solubility of CXB. In contrast to typical gelucires, Gelucire^®^ 44/14 and Gelucire^®^ 50/13, which are synthesized from lauroyl polyoxyl-32glycerides NF and stearoyl polyoxyl-32glycerides NF, respectively, Gelucire^®^ 48/16 is a unique type composed from PEG-32 stearate [13]. The hydrophobic interface between drug particles and the micelle core created by different Gelucire grades may vary depending on their chemical characteristics. In addition, it has been suggested that Gelucire^®^ 50/13 can be found in various crystalline forms that may affect its solubilizing abilities. Gelucire^®^ 44/14 has surface-active properties and a great capacity to self-emulsify during dissolving in a water-based medium to produce fine emulsions. Similar to Gelucire^®^ 48/16, which could effectively formulate micelle encapsulating hydrophobic drug when it comes into contact with an aqueous fluid [13].

The versatile behavior of different Gelucire^®^ grades could be attributed to differences in HLB values leading to the changeable solubilizing power of hydrophobic drugs. For example, the improvement in indomethacin solubility with Gelucire^®^ 48/16 aqueous solutions was more noticeable than with Gelucire^®^ 44/14 or Gelucire^®^ 50/13 aqueous solutions. This could be the result of Gelucire^®^ 48/16 having greater HLB values, which could improve the drug’s miscibility and dispersibility inside the carrier and increase its solubility [14].

Polyvinylpyllodine (PVP) has also been included in our study for many purposes. PVP is a highly safe, water-soluble polymer, biocompatible. In addition, it is very soluble in a variety of organic solvents, making it a common ingredient in many pharmaceutical formulation processes [15]. It is widely employed in electrospinning/electrospraying to formulate nano-delivery systems due to its ideal properties of spinnability. PVP has previously been used to encapsulate natural plant extracts such as green tea [16]. Additionally, PVP could be utilized as a solubilizer, promoting higher hydrophilicity in pharmaceutical formulations. PVP’s impact on improved bioavailability makes it an attractive polymer in newly developed controlled or targeted drug delivery systems [17].

In our study, we formulated CXB into GLR-PVP MPs to enhance its poor aqueous solubility to increase CXB’s oral bioavailability, which could be confirmed by in vivo anti-inflammatory in experimental rats.

## 2. Results

### 2.1. Solubility Study of CXB in Gelurice^®^ 48/16

Increasing GLR 48/16 concentration could significantly (*p* < 0.05) increase CXB’s solubility in 6% PVP aqueous solution, as displayed in Table 1. Starting with 6% PVP, the solubility of CXB was approximately 7 µg/mL and proportionally enhanced with increasing GLR concentration up to 25% *w*/*v* till the saturation point with no significant increase at 30% *w*/*v*. GLR concentration of 25% *w*/*v* was chosen to prepare the electrospraying solution. Also, values of Gibbs free energy change after the addition of Gelurice^®^ 48/16 have been calculated using the following equation [18].
(1)ΔGotr=−2.303 RT log SsS0
where *S*_s_/*S*_0_ is the ratio of molar solubility of CXB in 6% PVP aqueous solution with different concentrations of GLR 48/16 related to 6%PVP aqueous solution with no GLR. ΔG^o^_tr_ significantly (*p* < 0.05) increased with increased GLR 48/16 till the concentration of 25% *w*/*v*.

### 2.2. Characterization of GLR-PVP MPs Loaded with CXB

#### 2.2.1. Particle Size, PDI, Zeta Potential, Drug Loading (DL), and Surface Morphology

CXB-GLR-PVP MPs particle size average was 553.1 nm, and the PDI value was 0.217 (Figure 1A). The determined zeta potential was equal to −21.1 mV (Figure 1B). Drug loading results showed that each 10 mg of CXB-GLR-PVP MPs entraps approximately 1.98 ± 0.1 mg of CXB. Morphological examination using SEM (Figure 2) showed spherical CXB-GLR-PVP MPs with highly matched measurements of their diameters compared with the readings of dynamic light scattering (DLS).

#### 2.2.2. In Vitro Release Pattern

Regarding in vitro release behavior of CXB-GLR-PVP MPs, they showed a biphasic release pattern (Figure 3) starting with an initial burst at the first 10 min (% cumulative release equal to 50.31 ± 4.36) followed by a gradual increase of CXB mass released from microparticles till two hours. A significantly enhanced release (*p* < 0.01) of CXB from GLR-PVP polymeric matrix compared to free CXB. After 10 min, 50.3 ± 4.4% and 10.6 ± 2.8% have been cumulatively released from CXB-GLR-PVP MPs and free CXB, respectively.

The amount of released CXB increased significantly (*p* < 0.01) till reaching approximately 100% at 120 min, whereas free CXB exhibited a maximum cumulative % release of 33.6 ± 5.6 at the end of the experiment.

#### 2.2.3. FTIR Spectroscopy

IR peaks of CXB-GLR-PVP MPs, PVP K30, Gelurice^®^ 48/16, and free unprocessed CXB are shown in Figure 4. Table 2 shows the individual peaks of free CXB, PVP K30, and Gelurice^®^ 48/16. The spectrum of CXB-GLR-PVP MPs shows no extra bands with slight broadness observed at 3417 cm^−1^ (O-H stretching). Mostly, all the characteristic peaks of PVP K30 and Gelurice^®^ 48/16 are located at the same position in the spectrum of CXB-GLR-PVP MPs compared with their individual IR spectrums. Also, less intense peaks of CXB at 1162.60, 1347 cm^−1^ have been displayed in the IR spectrum of CXB-GLR-PVP MPs. The physical mixture prepared for IR analysis contained the amount of CXB, PVP K30, and Gelurice^®^ 48/16 mixed with their ratio equivalent to their amounts in CXB-GLR-PVP MPs. FT-IR spectrum of the physical mixture displayed no observed interaction between the components of our microparticle system. Additionally, the main characteristic peaks of CXB (S=O, 1348, 1164 cm^−1^) were shown at the same position compared with the individual spectrum of free CXB.

#### 2.2.4. X-ray Diffractometry (XRD)

XRD chart (Figure 5) demonstrates the peaks of pure unprocessed CXB, GLR, PVP, and CXB-GLR-PVP MPs to investigate the crystallinity of CXB after the electrospraying process. As shown in Figure 5, all the characteristic peaks of CXB completely disappeared after loading in GLR PVP aqueous solution. Also, the pattern of GLR-PVP MPs is highly similar to that of PVP plus two additional peaks at 2Ꝋ values of 19.52 and 23.7 (the same position of two distinct peaks showed on the GLR pattern). Regarding the XRD chart of the physical mixture, the same four distinguishing peaks of CXB are still observed with less intensity at 2Ꝋ values of 14.8, 16.1, 19.2, and 21.5.

### 2.3. In Vivo Anti-Inflammatory Study

#### 2.3.1. Edema Weight

As exposed in Figure 6, a remarkable decline (*p* < 0.05) in the edema weight was noticed in group V when compared with groups II, III, and IV. Group V revealed a marked decline with a percentage of 81.58% in relation to group II. In addition, it caused a marked diminish in relation to groups III and IV, with percentages of 65% and 63.61%, respectively.

#### 2.3.2. Histological Investigations

Paw sections of the studied experimental groups are revealed in Figure 7 and Figure 8 after staining with H&E and Masson’s trichrome stain, respectively.

#### 2.3.3. Immunohistochemical Investigations

The COX-2 and TNF-α immunostained paw tissue sections of the studied groups are revealed in Figure 9 and Figure 10.

#### 2.3.4. Pro-Inflammatory and Oxidative Stress Indicators

The influence of the placebo GLR PVP MPs, free CXB, and CXB-GLR-PVP MPs was investigated on PGE2 by ELISA (Figure 11) and NO as well as MDA (Figure 12) by colorimetric assay.

Group V revealed a considerable decline in PGE2 (11.11%) in relation to group II. Besides, it caused a marked decline in the level of PGE2 in relation to groups III and IV (13.04% and 5.88%, respectively).

Concerning the level of NO, in relation to group I, group II showed a noticeable rise with a percentage of 85.71%. Group V exposed a substantial reduction with a percentage of 37.69% in relation to group II.

Regarding the MDA level, in relation to group I, group II showed a noticeable rise (244.44%). Group V showed a marked lessening in MDA with a percentage of 67.74% in relation to group II.

#### 2.3.5. The mRNA Expression of IL-1β as Well as IL-6

The influence of placebo GLR PVP MPs, free CXB, and CXB-GLR-PVP MPs on IL-1β and IL-6 was elucidated by qRT-PCR, as revealed in Figure 13. In relation to group I, group II showed a significant rise in the expression of the IL-1β gene with a percentage of 130%. Group V showed a substantial decline, with a percentage of 52.17% in relation to group II.

Regarding IL-6 gene expression, in relation to group I, group II manifested a noticeable enhancement with a percentage of 140%. Group V manifested a marked decline with a percentage of 44.17% in relation to group II.

## 3. Discussion

Oral bioavailability is highly controlled by drug dissolution through GIT. So, aqueous solubility is a significant factor that should be fully considered, especially for Class II drugs. Celecoxib is a poorly soluble drug providing good stability at different pHs in GIT segments (pKa 11.1) [3]. The enhancement of aqueous solubility has been studied in several manuscripts using different formulation techniques such as solid dispersion [3,20], microemulsion [19], β-cyclodextrins’ complexation [21,22], physical manipulation of CXB’s solid state [23], etc. The reported aqueous solubility of CXB has been recorded to be 1.15 µg/mL. Additionally, PVP K30 has a crucial role in exaggerating CXB’s solubility and stability. It could efficiently entrap CXB into its matrix via hydrogen bonding formation producing a thermodynamically stable architecture of CXB in its amorphous state [24,25].

Gelurice^®^ 48/16 grade is a typical amphiphilic surfactant efficiently used for micellar solubilization. The enhancement of GXB’s solubility by Gelurice^®^ 48/16 could be attributed to additional wettability plus the micellar solubilization [13]. Values of Gibbs free energy change are an indication of the process of CXB molecules transfer into micelles offered by Gelurice^®^ 48/16. Increasing the negative value of ΔG^o^_tr_ means spontaneous solubilization of CXB that is enhanced with a higher concentration of GLR 48/16 till the saturation (Table 1) [18].

The electrospraying technique produces spherical microparticles (Figure 2) with a uniform size distribution and small PDI value (Figure 1A) [12]. Additionally, a perfect match was observed between the measurements of microparticles displayed on scanning electron microscopy and the results of dynamic light scattering. Also, the low value of PDI has been confirmed by SEM visualization. The particle size of CXB-GLR-PVP MPs mostly lies in the same ranges indicating the reproducibility of the electrospraying technique as the optimum method of preparation.

Regarding surface charge (Figure 1B), CXB-GLR-PVP MPs showed a well-established negative value equal to −20.21 ± 2.45 mV controlling optimum stability with little liability to flocculation or aggregation. The negative value might be attributed to the combinatory effect of PVP and GLR 48/18 [7,12,26]. The loading of CXB into the PVP matrix concurrently with micellar solubilization using GLR 48/16 could be an effective polymeric combination that efficiently entraps CXB leading to a higher drug loading capacity (0.198 mg of CXB per 10 mg of electrosprayed microparticles) [9,15].

The increased solubility of CXB in the presence of Gelurice^®^ 48/16 might be due to the drug’s increased wettability and micellar solubilization, which result in a high release profile of CXB-GLR-PVP MPs. The presence of Gelurice^®^ 48/16 in aqueous solutions could reduce the angle of contact between drug particles and water, improving drug particle wettability and, consequently, accelerating drug release. Additionally, the coexistence of hydrophilic carriers like Gelurice^®^ 48/16 may decrease the hydrophobicity of the drug particle surfaces, which will subsequently increase interaction with the aqueous medium and boost the rate of CXB’s dissolution [7]. Several previous studies showed the enhanced release performance exhibited after the addition of Gelurice^®^ 48/16 [27,28,29].

In our manuscript, the exaggerated release profile of CXB-GLR-PVP MPs could be attributed to more than one promoter. Firstly Gelurice^®^ 48/16 enhances CXB’s wettability and dispersibility via its surfactant properties. Secondly, the solubilizing effect offered by PVP, then ES, which has a stabilizing effect due to efficient drying keeping CXB in submicron level coated with a matrix composed of PVP and GLR [8].

Generally, the IR spectrum of CXB-GLR-PVP MPs indicates that no chemical interaction occurs during micellar solubilization and formulation of electrosprayed CXB-GLR-PVP MPs [9,30]. The broadness of OH peaks could be attributed to possible H- bonding formation between CXB and PVP background [15]. Additionally, less intense peaks of CXB might indicate the enclosed entrapment of CXB into the PVP matrix [15]. The peaks of CXB shown in the IR spectrum of the physical mixture indicate no possible interaction occurred with the active functional groups of PVP K30 and Gelurice^®^ 48/16. Generally, the overall fingerprint of the spectra of the physical mixture and CXB-GLR-PVP MPs are quite different, especially in the range between 2000 to 400 cm^−1^ [15]. Regarding the XRD chart, the disappearance of CXB’s peaks in the XRD chat ensures the presence of CXB in its amorphous soluble form [31]. A combined background of PVP and GLR has been displayed in the pattern of CXB-GLR-PVP MPs gathering the peaks of both PVP and GLR (2Ꝋ values of 19.52 and 23.7), indicating that PVP and GLR have contributed to the formation of polymeric microparticles encapsulating CXB [32]. The diffractogram of the physical mixture showed less intense peaks of CXB due to the low loading of weight per unit mass of the physical mixture compared with pure free CXB. The presence of the main peaks of CXB in the chart of the physical mixture indicates that CXB still exists in its crystalline form [32].

The inflammation process leads to a progression of various pathophysiological proceedings that induce the development of diseases [33]. Different pharmacological models performed on animals are used to elucidate the anti-inflammatory action of various compounds. A model usually utilized to estimate the anti-inflammatory activity of different drugs is the paw edema induced by carrageenan. Carrageenan is an agent that has the capacity to provoke the discharge of many pro-inflammatory in addition to inflammatory mediators [34].

Herein, group V or the treated group with CXB-GLR-PVP MPs exhibited a marked decline (*p* < 0.05) in the weight of the paw edema in relation to groups III or the treated group with free CXB and group IV or the treated group with placebo GLR PVP MPs. Edema is one of the important characteristics of the local inflammatory processes, which involves the increase of the interstitial fluids in the affected tissues [35]. Edema has a negative effect on tissue functions because these fluids make the diffusion of nutrients and oxygen more difficult. This could adversely affect the cellular metabolism in the affected tissue [36].

During the inflammation process, many cells, like neutrophils and macrophages, are stimulated to travel toward the site of the inflammation, generating huge amounts of mediators. Such substances are able to provoke an inflammatory response [37]. It is known that CXB has antipyretic, analgesic, and anti-inflammatory potential. Certain studies have been conducted to formulate CXB to improve its bioavailability and/or lessen its side effects [38,39,40]. The stimulation of the COX-2 pathway and the resulting enhanced discharge of PGE2 are major consequences of the acute inflammation of the carrageenan model [41]. In the current study, group II, or the positive control (carrageenan group), exhibited an enhancement of the COX-2 and TNF-α immunostaining (score 3). Interestingly, this finding was significantly lessened in the group treated with CXB-GLR-PVP MPs (group V). Besides, the histopathological analysis of the paw skin tissues showed that the CXB-GLR-PVP MPs treated group did not have any inflammation or edema, with a noticeable rise in the density of collagen fibers.

Regarding the inflammatory and pro-inflammatory biomarkers, CXB-GLR-PVP MPs revealed a marked decline (*p* < 0.05) in the PGE2, MDA, NO, IL-1β, and IL-6 (using ELISA and qRT-PCR) when compared to the free CXB. Comparable results were stated by El-Medany et al. [42] and Bazan et al. [39] in acetic acid-induced colon injury.

## 4. Material and Methods

### 4.1. Chemicals and Material

All the utilized reagents, chemicals, and solvents are of high-grade purity within the acceptable standard obtained from Merck, USA.

### 4.2. Solubility Study of CXB in Gelurice^®^ 48/16

A solubility study of CXB has been conducted according to Zhang et al. [43] with slight adaptation to determine the optimum concentration of GLR 48/16 in 6% PVP aqueous solution (the same composition of the electrosprayed polymeric solution for MPs fabrication). Variable concentrations of GLR 48/16 were prepared in a 6% PVP aqueous solution. An extra amount of CXB was added to GLR concentrations, followed by stirring overnight for 24 h at room temperature. After equilibrium, undissolved CXB was discarded after centrifugation. The samples were filtered by a membrane filter (0.45 μm). After proper dilution, CXB samples were assayed using a UV-Vis spectrophotometer (Evolution 300 spectrophotometer, Waltham, MA, USA) at 254 nm [44]. GLR 48/16 concentration which provides the highest CXB’s solubility, was selected to prepare CXB-GLR-PVP aqueous solution injected via electrospraying.

### 4.3. Electrospraying Process

Firstly, the polymeric solution composed of 6% PVP and 25% GLR loaded with CXB (50 mg) has been sprayed using NANON-01B, MECCO, LTD, Japan through a stainless steel needle provided with high voltage supply (28 KV). The spraying feed rate was adjusted to 0.1 mL/hr. The electrosprayed nozzles have been collected on a metallic drum coated with an aluminum sheet. The space separating the needle tip and the drum has been set at 10 cm [15]. After ten hours of electrospraying, CXB-GLR-PVP MPs were collected and kept in a desiccator for further in vitro and in vivo experimentation.

### 4.4. Characterization of GLR-PVP MPs Loaded with CXB

In our study, CXB-GLR-PVP MPs formulated via electrospraying have been characterized as particle size, uniformity of size distribution, zeta potential, drug loading, in vitro release platform, scanning electron microscopy, spectroscopic analysis by Fourier-Transform Infrared Spectroscopy (FTIR), and X-ray diffractometry (XRD) to ensure their suitability to be employed for in vivo anti-inflammatory study in experimental rats.

#### 4.4.1. Particle Diameter, Polydispersity Index (PDI), and Surface Charge

The average size of the particles (nm), uniformity of size distribution, and Zeta potential (mV) have been determined using Malvern Zetasizer Nano-zs90 (Malvern Instruments Ltd., Malvern, UK). A sample of CXB-GLR-PVP MPs (5 mg) was properly diluted with deionized water for precise analysis. All measurements have been performed in triplicate, expressed as mean ± SD [45].

#### 4.4.2. Drug Loading (DL)

An accurately weighted mass of CXB-GLR-PVP MPs was dissolved in dimethylformamide (DMF): ethanol with a ratio of 1:1 to study the amount of CXB loaded in the matrix of PVP-GLR. The sample was filtered, then assayed using a UV-Vis spectrophotometer, as previously mentioned in Section 4.2. DL was calculated as follows [45].
(2)$$\mathrm{DL}=\frac{Mass\;of\;CXB\;in\;MPS\;\left(mg\right)}{Mass\;of\;MPs\;\left(mg\right)}$$

#### 4.4.3. In Vitro Release

The accurate weight of CXB-GLR-PVP MPs (20 mg) has been immersed in a beaker of release medium (phosphate buffered saline, PBS, 100 mL, pH = 6.8) [8,15]. The release medium was agitated using magnetic stirring at 100 rpm and 37 °C. At different time periods till 2 h, samples of 2 mL were withdrawn for UV analysis. An equivalent fresh media was replenished, maintaining the total volume of the release medium at 100 mL. The samples were filtered prior to UV assay, as revealed in Section 4.2. Also, the release of free unprocessed CXB has been studied using the same previous procedures. All the release experiments have been implemented in triplicates, and the data presented as a mean of % cumulative release versus the time intervals till two hours.

#### 4.4.4. Scanning Electron Microscopy (SEM)

Scanning electron microphotographs of CXB-GLR-PVP MPs were visualized using JSM- 6510LV microscope (JEOL, Tokyo, Japan). The instrument was operated with an accelerated voltage equal to 30 kV. The sample was fixed on an aluminum carrier, and a sputter coater SPI-MODULE ^TM^ Sputter Coater (USA) was used to coat the sample with a gold film for 10 min. Measurements shown on SEM images were displayed using image J software [15].

#### 4.4.5. Spectroscopic Analysis by FTIR

It was employed to determine the possible interaction that could occur during micellar formation and the electrospraying process to prepare electrosprayed CXB-GLR-PVP MPs. FTIR spectra of unprocessed free CXB, GLR 48/16, PVPK30, and CXB-GLR-PVP MPs were documented in the 4000–400 cm^−1^ range by FTIR spectrophotometer (Bruker Tensor 27, Germany). Approximately three milligrams of each sample were blended with 200 mg of KBr (IR grade), followed by intense compaction into a spherical disc [46]. KBr-FTIR spectra have been developed at an optimum resolution of 4 cm^−1^ by a number of scans equal to 64. The acquisition parameters have been set at a smoothing of 17 and a base correction of 80.

#### 4.4.6. XRD

It is a significant investigation showing the presence of CXB in its crystalline or amorphous state after loading into PVP MPs. XRD was performed for unprocessed free CXB, GLR 48/16, PVP, and CXB-GLR-PVP MPs by APD 2000 PRO X-ray Diffractometer (GNR Instrumental Group, Italy). The X-ray diffractor was set at a current and source power equal to 25 mA and 35 kV, respectively, with 2Ꝋ angles ranging from 5 to 65 and a step size of 0.02 [47].

### 4.5. In Vivo Anti-Inflammatory Study

#### 4.5.1. Rats

Fifty Wistar albino male rats (their average weight 183.24 ± 8.23 g) were obtained. Their weights had a range of 180 to 210 g, and they were supplied with water (filtered) in addition to a diet of standard pellets. Our experimental protocol was credited by the Research Ethical Committee at the Faculty of Pharmacy, Tanta University, Egypt (TP/RE/10/22p-0053).

#### 4.5.2. Inflammation-Induced by Carrageenan

It was performed as previously described [30] (Appendix A).

#### 4.5.3. Protocol of the Experiment

Rats were grouped into five categories (n = 10). Normal control (group I) was administered orally with 0.9% saline. Positive control (group II) was inflamed by carrageenan injection and administered orally with 0.9% saline. Groups III, IV, and V were inflamed by carrageenan injection and administered orally placebo GLR-PVP-MPs, unprocessed free CXB, and CXB-GLR-PVP MPs, respectively (amount of unprocessed free CXB and loaded into CXB-GLR-PVP MPs was kept equivalent regarding rats’ weight). After six hours, all animals were anesthetized and euthanized. Then, we cut the left and right paws at nearly the same position, and their weights were identified. The paw edema average weight was determined by calculating the difference between the right and left paw weights [48].

#### 4.5.4. Histological Investigations

Tissues of the paws were conserved in a formalin solution, retained in paraffin wax, spliced, and stained using hematoxylin and eosin (H&E) and Masson’s trichrome stain for staining of collagen fibers [49].

#### 4.5.5. Immunohistochemical Studies

Using ABclonal Technology kits (Woburn, MA, USA), the COX-2 and TNF-α immune expression in the paw tissues was elucidated. The obtained pictures were analyzed, and they were given scores based on the percentage of the staining. The scores ranged from 0 to 3 regarding the positive staining percentages as previously described [50].

#### 4.5.6. ELISA and Colorimetric Assay

The quantity of prostaglandin E2 (PGE2) that are present in the tissues of paws in the different groups was determined by the ELISA kit obtained from Creative-Biolabs, New York, NY, USA. The absorbance was identified at 450 nm using a microtitration plate reader.

The amounts of nitric oxide (NO) and malondialdehyde (MDA) were determined by Biodiagnostic kits (Biodiagnostics, Egypt) at an absorbance of 540 nm.

#### 4.5.7. qRT-PCR

The inflammatory cytokines mRNA expression of IL-1β and IL-6 was studied in the paw tissues [51]. We used the beta-actin gene [52] as a housekeeping gene, and the utilized primers are presented in Appendix A. After extraction of the RNA using the Purelink™ RNA Mini Kit (Thermo Scientific, Waltham, MA, USA), RNA was turned into cDNA using the power™ cDNA synthesis kit. qRT-PCR was employed using Rotor-Gene Q 5plex (Qiagen, Hilden, Germany).

### 4.6. Statistics

Using Graph-Pad Software (prism 8), we performed ANOVA and post-hoc tests. Experiments were performed three times, and the obtained numbers are displayed as the mean ± standard deviation (SD). The significance was at *p* < 0.05.

## 5. Conclusions

Poorly soluble drugs could be efficiently entrapped into dry MPs after the utilization of carefully selected surfactant and hydrophilic polymer followed by ES. Gelurice® 48/16 has stabilized the system by improving the wettability concurrently with controlling the crystallinity of CXB. The presence of PVP potentiates the hydrophilicity of the aqueous dispersion and perfect electrospraying properties. Finally, the dried system with a submicron particle diameter of CXB completes the suitability of the micro delivery system to produce a significantly higher in vitro % cumulative release compared with unprocessed CXB. Hence, greater oral bioavailability could be expected. CXB-GLR-PVP MPs showed an improved anti-inflammatory capacity by causing a marked decline (*p* < 0.05) in the pro-inflammatory and inflammatory biomarkers in addition to decreasing the NO and MDA levels. These consequences occurred in substantial suppression of the COX-2 pathway.

## Figures and Tables

**Figure 1 pharmaceuticals-16-00258-f001:**
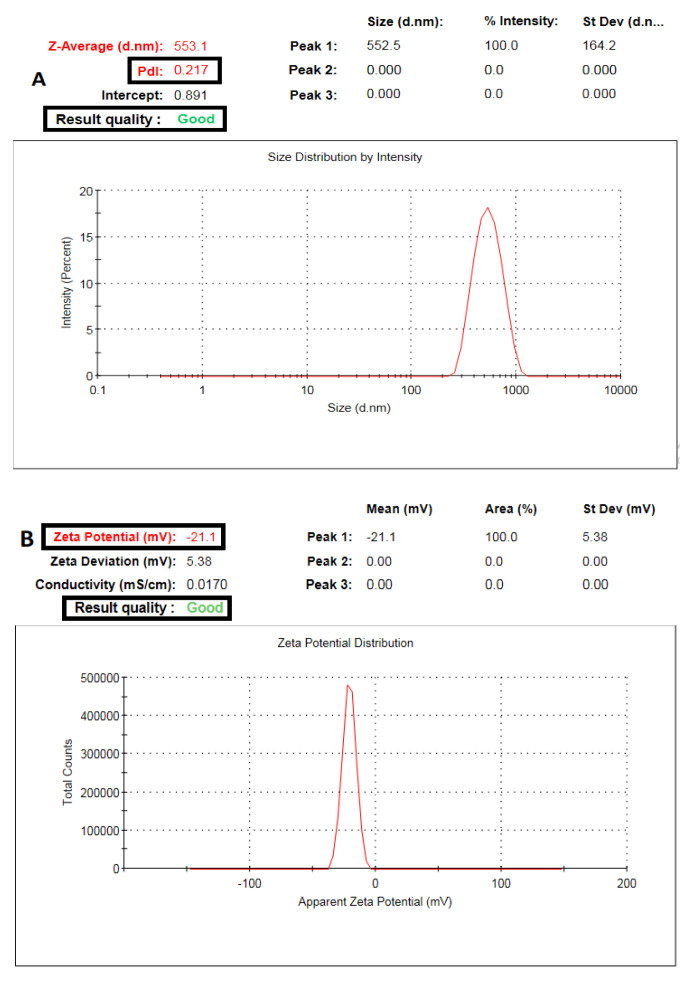
Malvern’s Zetasizer graphs of (**A**) Particle size and PDI, (**B**) ζ-potential values of CXB−GLR−PVP MPs exhibiting good quality results (Blacken outlined squares).

**Figure 2 pharmaceuticals-16-00258-f002:**
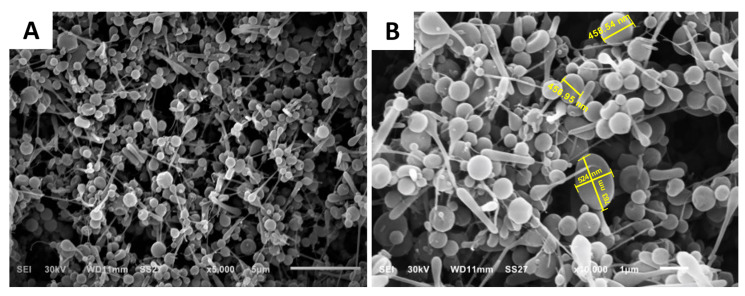
Scanning electron micrographs of CXB−GLR−PVP MPs were observed under (**A**) low magnification (×5000, scale bar 5 µm) and (**B**) high magnification (×10,000, scale bar 1µm).

**Figure 3 pharmaceuticals-16-00258-f003:**
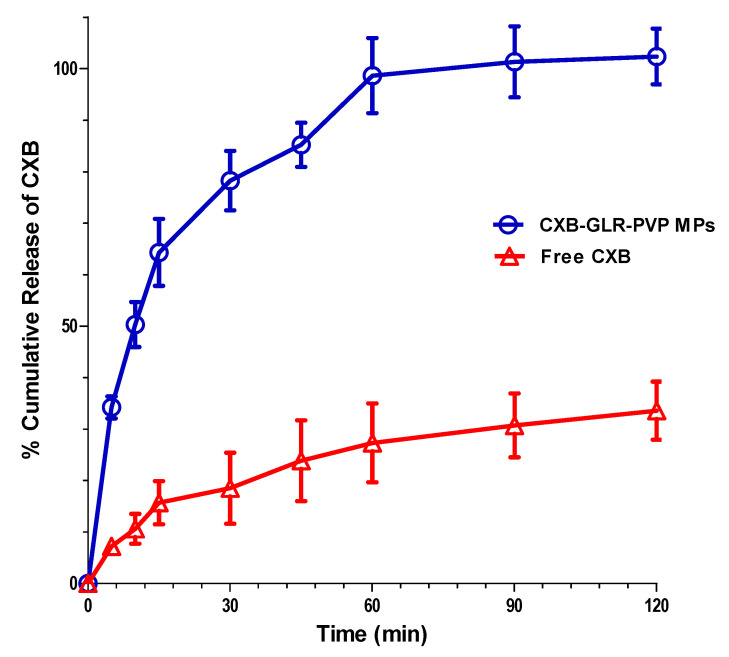
In vitro CXB release pattern of CXB-GLR-PVP MPs compared with unprocessed free CXB. All points are revealed as mean ± SD.

**Figure 4 pharmaceuticals-16-00258-f004:**
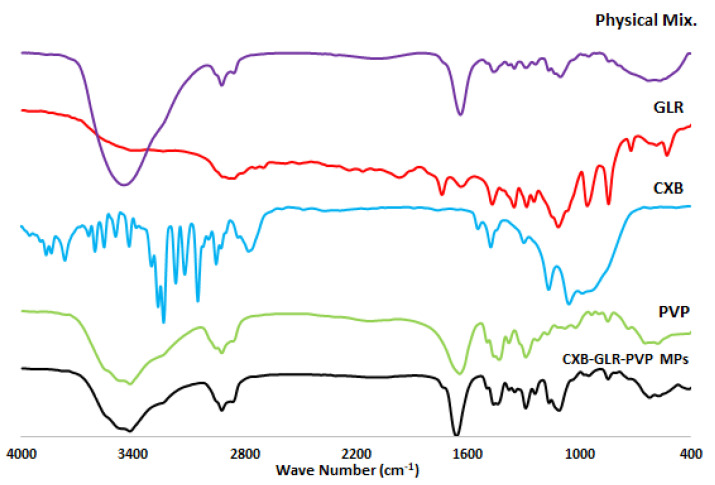
FTIR spectrum of CXB, Gelurice^®^ 48/16, PVP K30, physical mixture, and CXB-GLR-PVP MPs at a range of 400–4000 cm^−1^.

**Figure 5 pharmaceuticals-16-00258-f005:**
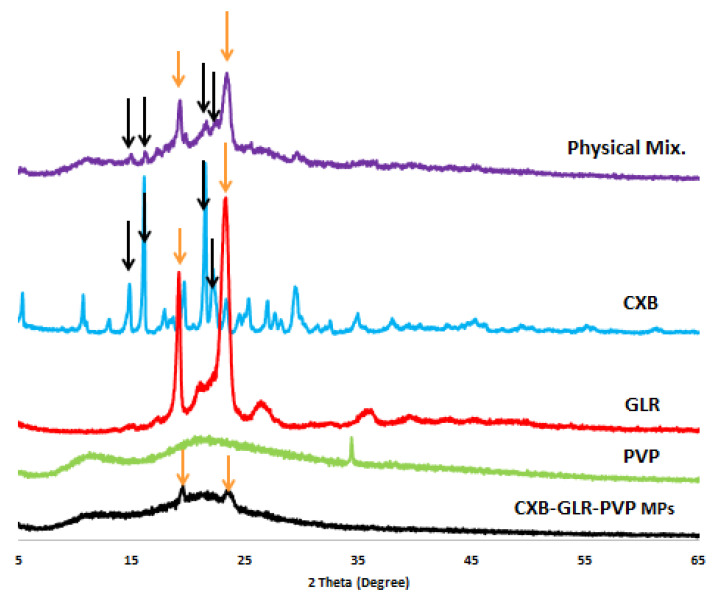
X-ray diffractogram of CXB, GLR, PVP, physical mixture, and CXB-GLR-PVP MPs showing the disappearance of sharp peaks of the CXB pattern. Orange arrows discriminate the characteristic peaks of GLR displayed on the pattern of pure Gelurice^®^ 48/16, physical mixture, and CXB-GLR-PVP MPs at the same position. Black arrows show the main peaks of CXB on the pattern of pure CXB and the physical mixture.

**Figure 6 pharmaceuticals-16-00258-f006:**
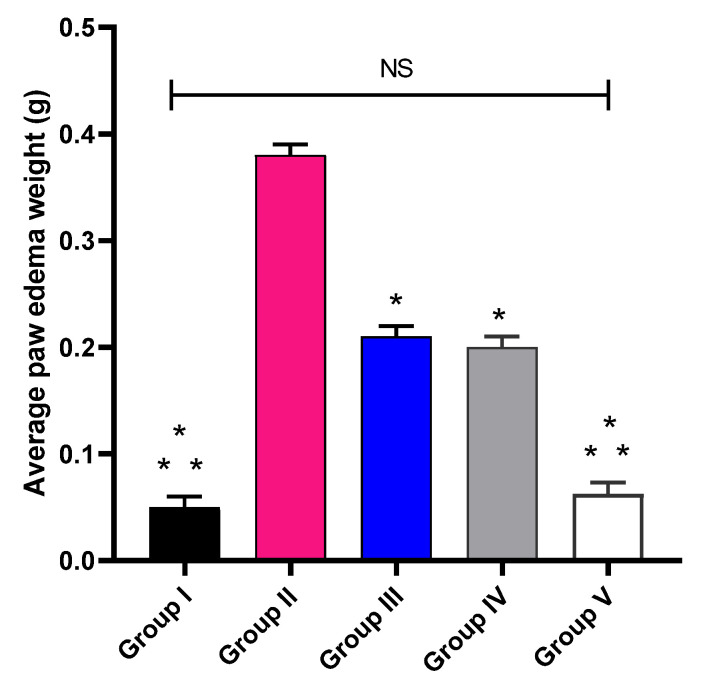
Average weight of paw edema. (*) designates a substantial difference (*p* < 0.05) in regard to group II. (**) designate a substantial variation (*p* < 0.05) in relation to groups III and IV. The NS acronym denotes a non-significant variation (*p* > 0.05).

**Figure 7 pharmaceuticals-16-00258-f007:**
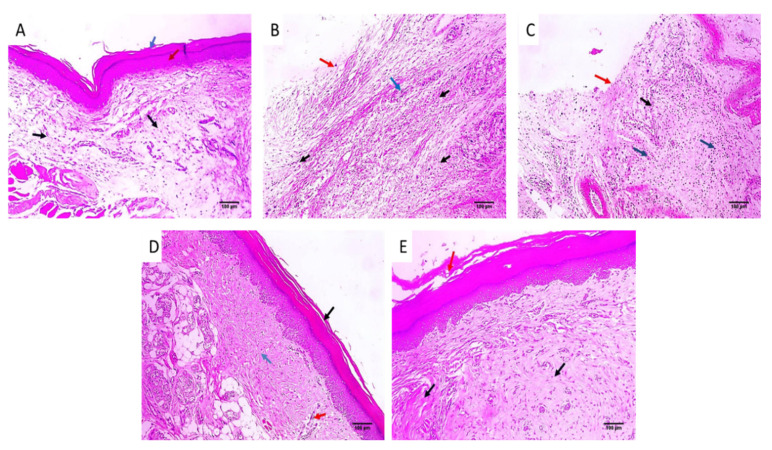
Paw skin sections stained with H&E of (**A**) Group I revealed normal skin. The epidermis has a normal thickness (red arrow) lined with keratin (blue arrow) and normal dermis (black arrows) below it (×100). (**B**) Group II and (**C**) Group III revealed surface ulceration (red arrow) with granulation tissue (blue arrow) with infiltration by inflammatory cells (black arrows) (×100). (**D**) Group IV revealed mild dermal inflammation (red arrow), and the epidermis had a higher thickness and mild keratosis (black arrow) with moderate collagenosis (blue arrow) (×100). (**E**) Group V revealed no inflammation, and the epidermis was thickened and covered with excessive keratosis (red arrow) with marked collagenosis (black arrow) (×100).

**Figure 8 pharmaceuticals-16-00258-f008:**
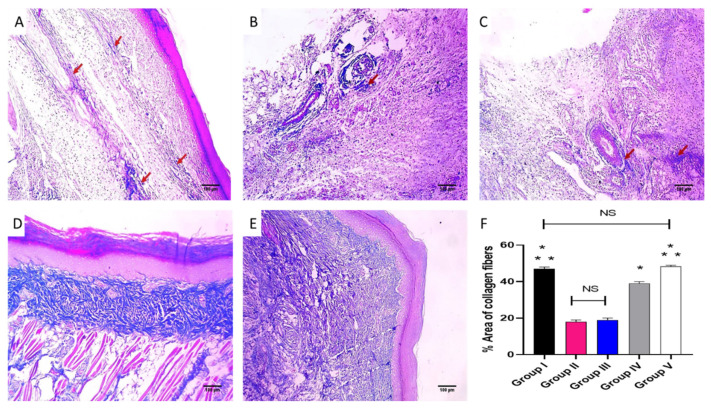
Paw skin sections stained with Masson’s trichrome stain of (**A**) Group I revealing bundles of collagen fibers (red arrows) (×100). (**B**) Group II and (**C**) Group III revealed focal collagen bundles (red arrows) (×100). (**D**) Group IV revealed a modest rise in the thickness of collagen (×100). (**E**) Group V revealed a noticeable rise in the thickness of collagen (×100). (**F**). The bar5 chart showed collagen fiber percentage. (*) designates a substantial variation (*p* < 0.05) in relation to group II. (**) designate a substantial variation (*p* < 0.05) with regard to groups III and IV. (NS) designates a non-significant variation (*p* > 0.05).

**Figure 9 pharmaceuticals-16-00258-f009:**
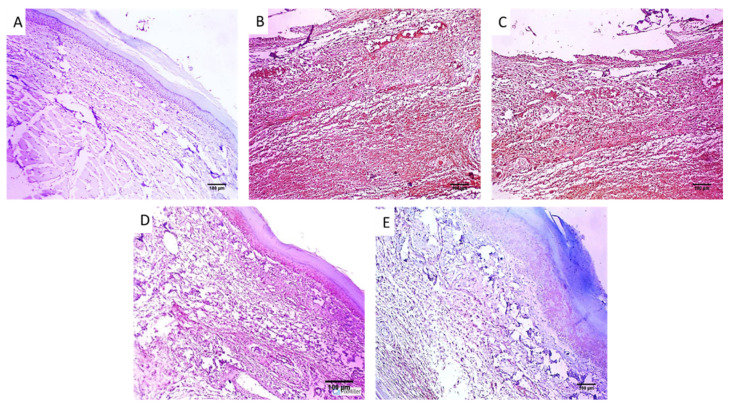
Paw skin COX-2 immunostained segments of (**A**) Group I revealed negative COX-2 immunostaining having a score of 0 (×100). (**B**) Group II and (**C**) Group III revealed a strong positive COX-2 immunostaining having a score of 3 (×100). (**D**) Group IV revealed a moderate positive COX-2 immunostaining having a score of 2 (×100). (**E**) Group V revealed a mild positive COX-2 immunostaining having a score of 1 (×100).

**Figure 10 pharmaceuticals-16-00258-f010:**
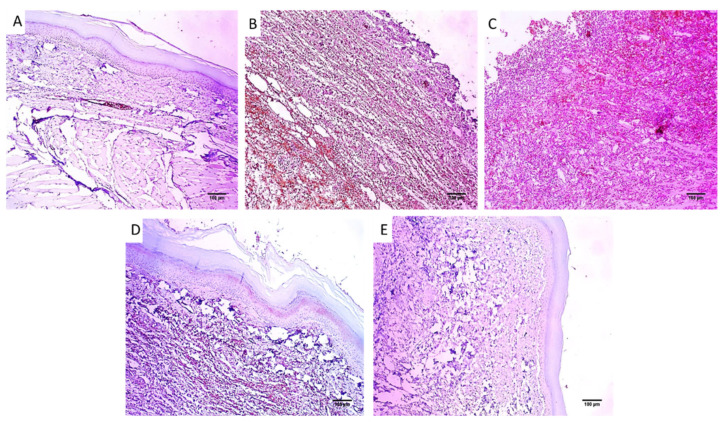
Paw skin TNF-α immunostained segments of (**A**) Group I and (**E**) Group V revealed negative TNF-α immunostaining, having a score of 0 (×100). (**B**) Group II and (**C**) Group III revealed strong positive TNF-α immunostaining, having a score of 3 (×100). (**D**) Group IV revealed a moderate positive TNF-α immunostaining having a score of 2 (×100).

**Figure 11 pharmaceuticals-16-00258-f011:**
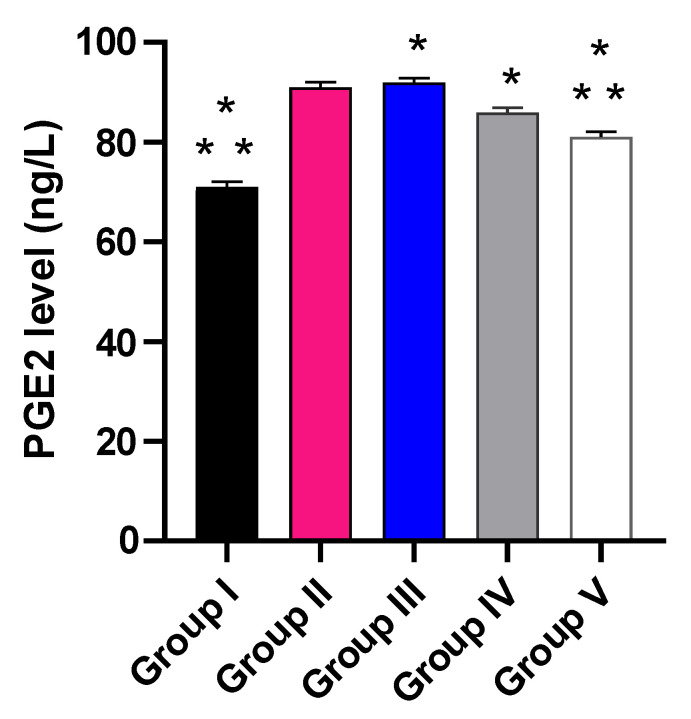
A graph displaying the levels of PGE2 in the different studied groups. (*) designates a substantial variation (*p* < 0.05) in relation to group II. (**) designate a substantial variation (*p* < 0.05) in relation to groups III and IV.

**Figure 12 pharmaceuticals-16-00258-f012:**
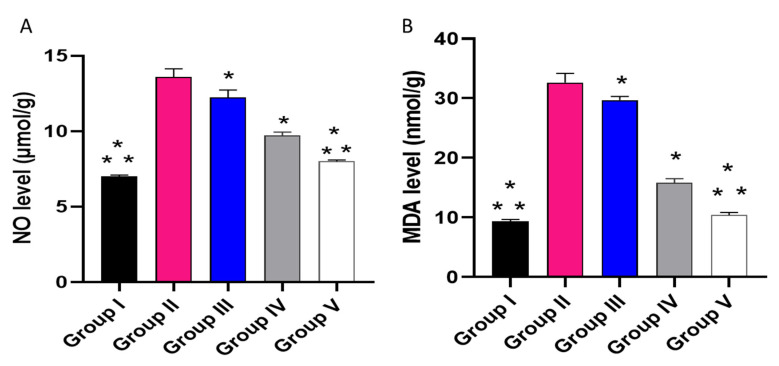
A graph displaying (**A**) NO and (**B**) MDA in the different studied groups. (*) designates a substantial variation (*p* < 0.05) in relation to group II. (**) designate a substantial variation (*p* < 0.05) in relation to groups III and IV.

**Figure 13 pharmaceuticals-16-00258-f013:**
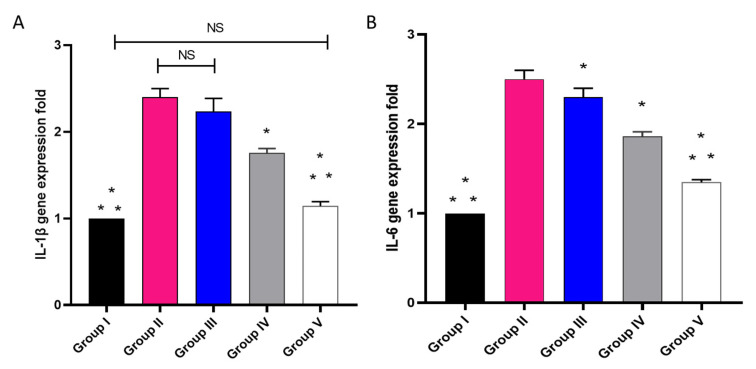
A graph displaying mRNA expression of (**A**) IL-1β and (**B**) IL-6 in different studied groups. (*) designates a substantial variation (*p* < 0.05) in relation to group II. (**) designate a considerable variation (*p* < 0.05) with regard to groups III and IV. (NS) designates a non-significant variation (*p* > 0.05).

**Table 1 pharmaceuticals-16-00258-t001:** Solubility and thermodynamic properties of CXB in 6% PVP aqueous solution at 25 °C.

Gelurice^®^ 48/16 Conc. (% *w*/*v*)	Solubility of CXB (mg/mL)	ΔG^o^_tr_ (kJ/moL)
0	0.007 ± 0.001	0.00
5	0.752 ± 0.025	−9.52
10	1.256 ± 0.029	−11.26
15	2.136 ± 0.036	−12.45
20	4.359 ± 0.031	−14.36
25	7.236 ± 0.039	−15.69
30	7.934 ± 0.048	−15.89

**Table 2 pharmaceuticals-16-00258-t002:** Characteristic IR peaks of PVP K30, Gelurice^®^ 48/16, and free unprocessed CXB.

	Assignment	Approximate Frequency (cm^−1^)
CXB [11,19]	NH	3344, 3235
	S=O	1348, 1164
	NH	1621
	aromatic CH	761
PVP K30 [10]	O-H	3416
	-CH2	2953, 2922
	C=O	1642
	C-H	1378
	C-N	1287
	δ CH2	846
	δ N-C=O	577
Gelurice^®^ 48/16 [9]	sulfone	1112
	C=O	1738, 1636
	=C-H & =CH_2_	956, 842
	O-H	721
	C-N	1243
	O-H	2858

## Data Availability

Data is contained within the article and Appendix A.

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
