# Peer review of "Fabrication of Celecoxib PVP Microparticles Stabilized by Gelucire 48/16 via Electrospraying for Enhanced Anti-Inflammatory Action"

_pharmaceuticals, 2023, doi:10.3390/ph16020258_

Round 1
Reviewer 1 Report
Overall, the manuscript is well-written and showed an efficient strategy for enhancing the solubility of CXB via ES and micellar solubilization followed by in vivo anti-inflammatory study.
Below are my comments and concerns for improvement:
Regarding the formulation of CXB-GLR-PVP MPs
- GLR has different grades and types, the rationale for your choice of GLR 48/16
- You studied the solubility of CXB using different concentrations of GLR in 6 % PVP, the solubility study should be first initiated in the water!
- Did you perform any optimization of the ES process to adjust the feed rate and applied voltage??
Regarding in vivo anti-inflammatory study
- Please add more details in the qRT-PCR method
- Why did you use the carrageenan-induced paw edema model in your study
- the role/rationale of Masson staining in your study needs more clarification.
Author Response
Many thanks for your valuable comments. I really appreciate your considerable time and effort.
Please see attached file

Author Response

(The authors gave the same response as above.)

Reviewer 3 Report
The manuscript of Alshawwa et al. on Celecoxib/PVP nanoparticles is well organized and correctly presents the results. Unfortunately, due to many minor shortcomings, it needs a major revision before its appearance.
- Inconsistent character formats are in the following lines:
138-140, 142, 148-149, 269, 278-280, 281, 283, 405-406, 487-492
- Section 2.4.5 is incomplete. The acquisition parameters (resolution, number of scans, data interval) and the method of handling the spectra (smoothing, baseline correction, possible normalization wavenumber, etc.) are missing.
- ΔG0tr is included in Table 1 but defined later. Please also consider using 0 (zero) in the last column of row 1, as the differences are for the 0% Gelurice. Otherwise, in this case, S0/Ss=1, i.e., ΔG0tr=0.
On the other hand, the referee assumes that Gelurice addition results in Ss>S0, i.e., S0/Ss<1, so the log (S0/Ss) is negative. Because negative*negative => positive, something might be incorrect in Table 1 or eq. 2. (in line 402, q. is seen as ΔG0tr= -2.303 RT log ?0/??).
- In section 3.2.1, the numbers in the text and figures are inconsistent. Different numbers are in Figure 1a/1b and text.
On the other hand, the two decimal places at ~600 nm are uninterpretable. Similarly, since zeta potential measurements have a relatively high experimental error and association property characterization limits are wide, two decimal places seem exaggerated. The three decimals digits of the calculated deviation for the encapsulated CXB have little meaning. Please use rounding to reduce decimal places.
- The identification of Figures 2a and 2b is missing from the images. On the other hand, the largest ellipsoid in the assumed Figure 2b shows the minor-axis value of the largest particle. Please also include the major-axis value.
- In lines 261 and 264, the two decimal places of the cumulative emission percentages are uninterpretable. The reviewer suggests rewording the sentence and adding '%' after the numeric values.
- In section 3.2.3, the IR bands should be tabulated. On the other hand, assuming that the authors used a resolution of 4 cm-1, the two decimal places in the IR bands do not provide additional information or greater precision.
- The physical mixture spectra should be included in the IR and XRD figures, as is usual for papers on similar topics.
- Journal abbreviations should include points in the References section. In addition, DOIs are also missing.
Author Response
Many thanks for your valuable comments. I really appreciate your considerable time and effort.
Please find the attached file

Round 2
Reviewer 1 Report
I must commend the authors for addressing all the comments and concerns raised by me.
I don't have any further comments.
Author Response
Great thanks for your valuable comments.
Kind Regards
Dr. Dalia H. Abdelkader
Assistant Professor of Pharmaceutical Technology,
Head of Pharmaceutical Service Centre.
Faculty of Pharmacy, Tanta University, Egypt.
Reviewer 3 Report
Although the authors significantly improved the quality of their manuscript, two issues remained uncorrected, so a second major revision is necessary. The authors gave correct feedback for the referee's concerns, but the last two are difficult to accept. The reviewer also challenges the authors' response to the different values in Figure 1a/b and the text.
The S0/Ss problem is also confusing because the S0 usually means the solubility in a solvent without any manipulations, though the answer is more or less acceptable.
- The referee understands that the spectra of physical mixtures do not give new information to the authors. When a complex composition is studied, the spectrum registration of the individual components and physical mixture(s) (in identical weight ratio) is mandatory. Please include the missing spectra (IR and XRD).
- It might be true that the referee has missed something, but in the published articles, the journal abbreviations contain dots.
See, e.g., a recent article (https://www.mdpi.com/1999-4923/15/2/323), where the readers find the following line:
2. Nowaczyk, A.; Kowalska, M.; Nowaczyk, J.; Grześk, G. Carbon Monoxide and Nitric Oxide as Examples of the Youngest Class of Transmitters. Int. J. Mol. Sci. 2021, 22, 6029. [Google Scholar] [CrossRef].
The feedback to the remark 'In section 3.2.1, the numbers in the text and figures are inconsistent. Different numbers are in Figure 1a/1b and text.'
is
'The number mentioned in the text is the average ±SD for triple measurements whereas the figures displayed are just a representative of only one measurement.'
The referee suggests including figures of all measured data and removing the "representative example" from the main text to resolve the inconsistency. The pictures/figures are usually more informative than the text, and the different values are hard to understand.
Author Response
Great thanks for your valuable comments.
Dr. Dalia H. Abdelkader
Assistant Professor of Pharmaceutical Technology,
Head of Pharmaceutical Service Centre.
Faculty of Pharmacy, Tanta University, Egypt.

Round 3
Reviewer 3 Report
The repeated revision further improved Alshawwa and colleagues' manuscript on celecoxib PVP microparticles and turned it into a good article. It is now suitable for publication.
The reviewer's final request is to insert missing figures of particle sizes, as mentioned at the end of their feedback, and upload the updated SI, as that file is not size-limited.